# Magnetic Phase Diagram of van der Waals Antiferromagnet TbTe_3_

**DOI:** 10.3390/ma15248772

**Published:** 2022-12-08

**Authors:** Olga S. Volkova, Abdellali Hadj-Azzem, Gyorgy Remenyi, Jose Emilio Lorenzo, Pierre Monceau, Alexander A. Sinchenko, Alexander N. Vasiliev

**Affiliations:** 1Department of Low Temperature Physics and Superconductivity, Physics Faculty, Lomonosov Moscow State University, 119991 Moscow, Russia; 2Laboratory of Quantum Functional Materials, National University of Science and Technology “MISiS”, 119049 Moscow, Russia; 3CNRS, Grenoble INP, Institut NEEL, Université Grenoble Alpes, 38042 Grenoble, France; 4Kotelnikov Institute of Radioengineering and Electronics of RAS, 125009 Moscow, Russia; 5Laboratoire de Physique des Solides, Universite Paris-Saclay, 91405 Orsay, France

**Keywords:** terbium tritelluride, thermodynamics, transport properties, magnetic phase diagram

## Abstract

Terbium tritelluride, TbTe_3_, orders antiferromagnetically in three steps at *T_N_*_1_ = 6.7 K, *T_N_*_2_ = 5.7 K, and *T_N_*_3_ = 5.4 K, preceded by a correlation hump in magnetic susceptibility at *T** ~8 K. Combining thermodynamic, i.e., specific heat *C_p_* and magnetization *M*, and transport, i.e., resistance *R*, measurements we established the boundaries of two commensurate and one charge density wave modulated phases in a magnetic field oriented along principal crystallographic axes. Based on these measurements, the magnetic phase diagrams of TbTe_3_ at *H*‖*a*, *H*‖*b* and *H*‖*c* were constructed.

## 1. Introduction

Transition metal-based van der Waals compounds attract attention in the field of spintronics since they represent a natural crystal-perfect alternative to quasi-two-dimensional magnetic films obtained by epitaxial methods [1]. Also, they are of interest in fundamental research, providing a new platform for the study of low-dimensional magnetism. Recently, the halides and chalcogenides of rare-earths *RE* have emerged as important objects to reveal the effects of magnetic anisotropy in the processes of the long-range order formation in van der Waals compounds.

The layered YbCl_3_ can be considered as a model van der Waals system to investigate Heisenberg interactions on a honeycomb net. It has been shown that this material reaches the Neel-type ground state with a reduced moment and anisotropic in-plane bond-dependent coupling preceded by a short-range magnetic order. It satisfies the prerequisites of the Kitaev-Heisenberg model, lying on the border of Kitaev spin liquid from the antiferromagnetic side [2]. The rare-earth chalcohalides also belong to the van der Waals class of Kitaev spin liquid candidates. YbOCl has been investigated numerically in this respect, evidencing an exclusively rich magnetic phase diagram in terms of anisotropic exchange interactions *J*_±_ and *J*_zz_. Along with Neel, collinear, stripy, zigzag, and 120-AFM phases, this diagram hosts the spin-disordered one related presumably to the Kitaev physics [3]. 

The rare-earth chalcohalides with large magnetic moments and Ising-type anisotropy behave differently compared to Yb compounds with effective moment *J*_eff_ = 1/2. Thus, DyOCl reaches Neel order at low temperatures with magnetic moments oriented along the easy axis in the plane of the van der Waals layer. A moderate external field applied along this axis generates a spin-flip transition transforming DyOCl into a ferromagnet [4]. 

The rare-earth tellurides *RE*Te_3_ are known as a family of van der Waals compounds hosting the charge density waves and experiencing antiferromagnetic order at low temperatures [5]. Recently, the effect of a magnetic field on thermodynamic properties in GdTe_3_ has been investigated. Two antiferromagnetic transitions at *T_N_*_1_ = 11.5 K and *T_N_*_2_ = 9.7 K and an anomaly at *T*_1_ = 7 K were found in both magnetic susceptibility *χ* and specific heat *C_p_*. It was conjectured that the anomaly at *T*_1_ might originate from the incommensurate charge density wave in the van der Waals coupled layers of GdTe_3_ [6].

Various aspects of charge density wave transitions in *RE* tritellurides are discussed in [7]. Within this family, TbTe_3_ stands out as material evidencing three successive magnetic phase transitions at *T_N_*_1_ = 6.6 K, *T_N_*_2_ = 5.6 K, and *T_N_*_3_ = 5.4 K ascribed to the interplay of magnetism with charge density waves [8]. It possesses an orthorhombic structure described by space group *Cmcm* with lattice parameters *a* = 4.298 Å, *b* = 25.33 Å, and *c* = 4.303 Å at 300 K [9]. A slight difference in *a* and *c* lattice constants originates from the formation of a charge density wave with propagation vector *q_c_* = (0, 0, 0.296) at *T_c_* = 330 K [10]. It contains Tb-Te bilayers alternating with double tellurium layers Te^−1/2^ along the *b*-axis shown in the left panel of Figure 1. The Tb-Te bilayer constitutes a square lattice with Tb-Tb distances along the rungs *d_r_* ~5.1 Å and along the diagonals *d_d_* ~4.3 Å, as shown in the right upper and lower panels of Figure 1. Competing exchange interactions between rare earth magnetic dipoles are *J_d_*_1_ ≈ *J_d_*_2_ ≫ *J_r_* due to the difference in distances. In accordance with neutron diffraction data [9], the ordering vectors of antiferromagnetic structures below *T_N_*_1_ (AF1) and *T_N_*_2_ (AF2) are equal to *q_m_*_1_ = (1/2, 1/2, 0) and *q_m_*_2_ = (0, 0, 1/2). The deviations of these structures from exact commensurability are discussed in Ref. [10]. Herein, we represent the study of transport, magnetic, and thermal properties of TbTe_3_ aimed at the determination of magnetic phase diagrams along principal crystallographic axes.

## 2. Experimental

Single crystals of TbTe_3_ were grown by a self-flux technique under a purified argon atmosphere, as described previously [11]. High-quality single crystals were selected for the measurements. Thin plates with a thickness of less than 1 μm were prepared by micromechanical exfoliation from thick plates glued on a sapphire substrate. The quality of selected crystals and the spatial arrangement of crystallographic axes were controlled by X-ray diffraction techniques.

The bridges with a width of 50–80 μm in well-defined, namely (100) and (001), orientations were cut from un-twinned single crystals. Measurements of temperature dependencies of resistance have been performed with a conventional 4-probe configuration. Gold evaporation and indium cold soldering were employed for the preparation of electric contacts. Measurements in the magnetic field were done in superconducting solenoid up to B = 8 T.

Thermodynamic measurements, i.e., magnetization and specific heat, were performed on the thin plates of square shape weighing several mg. Measurements for *B*‖*b* were done using the vibrating sample magnetometer option for Physical Properties Measurement System PPMS 9T “Quantum Design”. Measurements for *B*‖*ac* were done by means of Magnetic Properties Measurement System MPMS 7T “Quantum Design”.

## 3. Thermodynamic Properties

At elevated temperatures, *M*/*H* vs. *T* curve in TbTe_3_ follows the Curie-Weiss law *χ* = *χ*_0_ + *C*/(*T* − *Θ*) with temperature independent term *χ*_0_ = − 5 × 10^−4^ emu/mol, Weiss temperature *Θ* = −3.5 K, and Curie constant *C* = 12.3 emu K/mol. The value of the Curie constant allows estimating the effective magnetic moment of Tb^3+^ ions according to the ratio 8 *C* = *µ_eff_*^2^. It gives the *µ_eff_* = 9.92 µB, which is in good correspondence with the tabular magnetic moment of Tb^3+^ ions *µ_calc_* = 9.7 µB. At lowering temperatures, the χ(*T*) curve deviates downward from the extrapolation of the Curie-Weiss law and evidences a round hump at *T** ~8 K. This feature is ascribed routinely to correlation effects in low dimensional magnetic systems [12]. Specific heat *C_p_* in TbTe_3_ evidences sharp singularities at *T* < *T**. The temperature dependences of both magnetization *M* and specific heat *C_p_* are shown in the left panel of Figure 2. The close inspection of the low-temperature portions of *M*/*H* vs. *T* and *C_p_* vs. *T* curves allows the revealing of these singularities. As shown in the right panel of Figure 2, there are three features on *M*/*H* curves at *T_N_*_1_ = 6.7 K, *T_N_*_2_ = 5.7 K, and *T_N_*_3_ = 5.3 K most pronounced in the Fisher specific heat *d[(M/H)T]/dT*. This function is closely similar to the magnetic specific heat in the region of transition [13]. These three anomalies are also seen in the *C_p_* vs. *T* curve, which mainly consists of the contribution from the magnetic subsystem in the region of transition sharpest of them is detected at *T_N_*_2_.

Selected temperature dependences of *M*/*H* taken along principal crystallographic axes in TbTe_3_ at various magnetic fields are shown in Figure 3. Full measurements at *H*//*b* were possible only at the lowest magnetic field *µ*_0_*H* = 0.01 T; at higher fields, the field-induced mechanical readjustment of the sample prevented reliable measurements. At increasing magnetic field, the *M*/*H* vs. *T* curves monotonically shift to lower temperatures, essentially keeping their appearance. There are two sharp peaks at *T_N_*_3_ and *T_N_*_2_ and a shoulder at *T_N1_* on all these curves, corresponding to the succession of three magnetic phase transitions.

The field dependences of magnetization taken at various temperatures along principal axes of TbTe_3_ are shown in Figure 4. *M*(*H*) curves measured along the *a* and *c* axes look identical. Both curves are close to zero in weak magnetic fields and evidence spin-flip transition at *µ*_0_*H* = 2.33 T followed by the tendency to saturation. *M*(*H*) curve measured along the *b*-axis grows almost linear in the external magnetic field and evidences a sequence of small jumps at *µ*_0_*H_C_*_3_ = 2.9 T, *µ*_0_*H_C_*_2_ = 3.8 T, and *µ*_0_*H_C_*_1_ = 5.9 T attributed to the crossings of the phase boundaries. These data allow suggesting that the magnetic moments of Tb^3+^ ions are oriented within the *ac* plane.

## 4. Transport Properties

Temperature dependences of in-plain resistance at various orientations of the external magnetic field with respect to crystal axes and electric current *I* are shown in Figure 5. At the lowest magnetic fields, all *R*(*T*) dependences show a drop between *T*_N3_ and *T*_N2_. This tendency is gradually suppressed by the magnetic field. The kinks at *R*(*T*) curves marked by arrows correspond to the phase transitions identified in thermodynamic measurements. This means that, despite the assumption that the electric current *I* presumably flows through the double layers of tellurium, the processes occurring in the terbium—tellurium bilayers have a significant effect on the scattering of current carriers. Note that the effect of normal carriers scattering on magnetic orders depends strongly on the direction of the applied electric field (see Figure 5a,b) and becomes much more pronounced under the application of a magnetic field. As can be seen, the peculiarity corresponding to the AF1 transition appears in *R(T)* curves only in the magnetic field and in the form of an increase of resistance in contrast to results of [5], where all three transitions have been identified only from derivative *dR(T)/dT* and appear as decrease of resistance.

## 5. Magnetic Phase Diagram

To establish the boundaries of various magnetic phases in TbTe_3_ in a magnetic field oriented along principal crystallographic axes, the *dM/dT*, *dM/dH,* and *dR/dT* derivatives were analyzed. The crossings of phase boundaries were identified as the sequences of peaks. The positions of these peaks are shown in the magnetic field—temperature phase diagrams, as shown in Figure 6. Additionally, sometimes unpronounced features are shown by dash lines. Overall, the data obtained in thermodynamic and transport properties agree well and complement each other.

## 6. Summary

The boundaries of two commensurate AF1 at *T_N_*_1_ and AF2 at *T_N_*_2_ and incommensurate AF3 at *T_N_*_3_ phases, as defined in neutron scattering measurements in the absence of an external magnetic field, are confirmed in thermodynamic and transport measurements. The application of magnetic field shifts these boundaries to lower temperatures in parallel. In thermodynamic properties, the anomalies *T_N_*_3_ and *T_N_*_2_ are seen as the sharp signatures of the first-order transitions. It can be due to the giant magnetostriction of the rare-earth compounds, which transforms λ-type anomaly inherent for a second-order phase transition of purely magnetic origin to a symmetric sharp peak at the magnetostructural phase transition. At the same time, the anomaly at *T_N_*_1_ is seen as a shoulder in both *C_p_*(*T*) and Fisher heat. It is not always straightforward to detect this anomaly under a magnetic field, which usually smears the transitions. In essence, the magnetic phase diagram obtained provides a full description of the magnetic properties of the terbium tritelluride.

## Figures and Tables

**Figure 1 materials-15-08772-f001:**
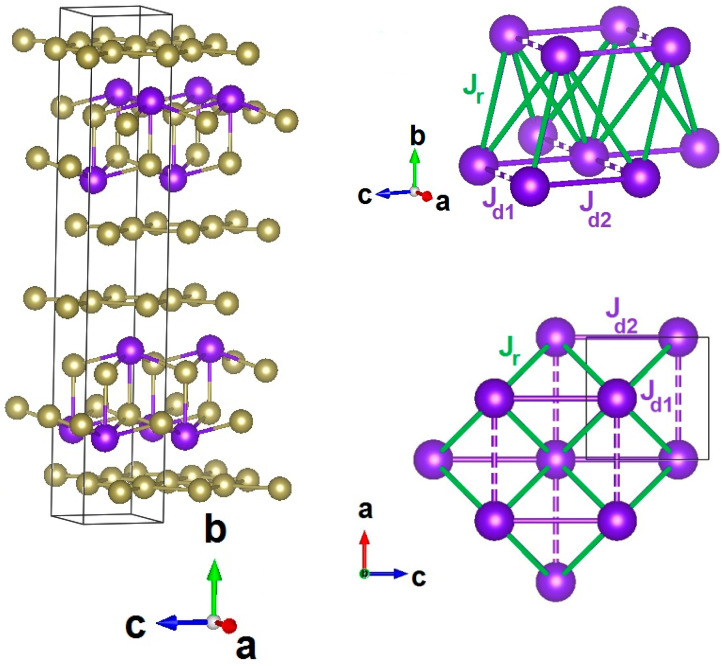
(**Left panel**): Crystal structure of TbTe_3_. Large and small spheres represent Tb and Te atoms. (**Upper** and **Lower right panels**): Three-dimensional net and *ac* projection of corrugated Tb—Te bilayer. Green, blue, solid, and dotted lines denote dipolar magnetic exchange interaction pathways along the rungs *J_r_* and diagonals *J_d_*_1_ and *J_d_*_2_.

**Figure 2 materials-15-08772-f002:**
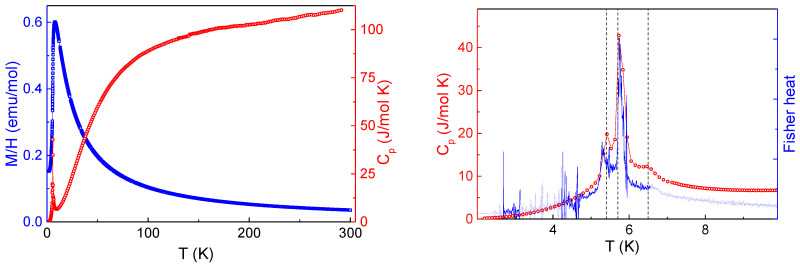
**(Left panel**): Temperature dependences of specific heat *C_p_* and magnetic susceptibility χ = *M*/*H* at *µ*_0_*H* = 0.1 T in TbTe_3_. (**Right panel**): Fisher specific heat *d[(M/H)T]/dT* vs. *T* and *C_p_* vs. *T* enlarged. Vertical dash lines denote temperatures of magnetic phase transitions at *T_N_*_1_, *T_N_*_2,_ and *T_N_*_3_.

**Figure 3 materials-15-08772-f003:**
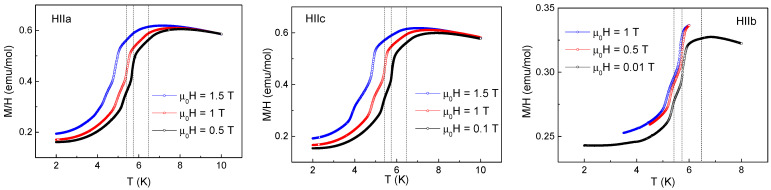
Temperature dependences of *M*/*H* taken at various external magnetic fields along principal crystallographic axes in TbTe_3_. Vertical dash lines denote temperatures of magnetic phase transitions at *T*_N1_, *T*_N2,_ and *T*_N3_.

**Figure 4 materials-15-08772-f004:**
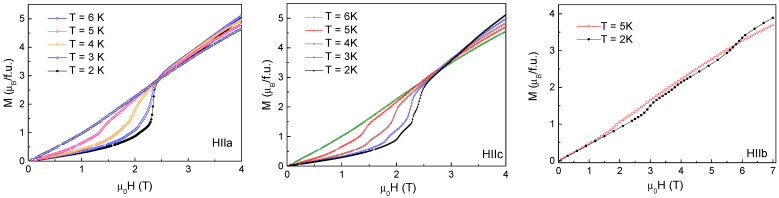
The field dependences of magnetization taken at various temperatures along principal axes in TbTe_3_.

**Figure 5 materials-15-08772-f005:**
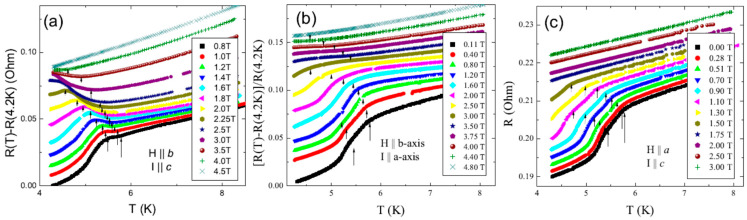
Temperature dependences of resistance *R* at various orientations of magnetic field with respect to crystal axes and electric current: (**a**) *H*‖*b* and *I*‖*c*; (**b**) *H*‖*b* and *I*‖*a*. (**c**) *H*‖*a* and *I*‖*c*. The curves obtained at various fields are shifted for clarity.

**Figure 6 materials-15-08772-f006:**
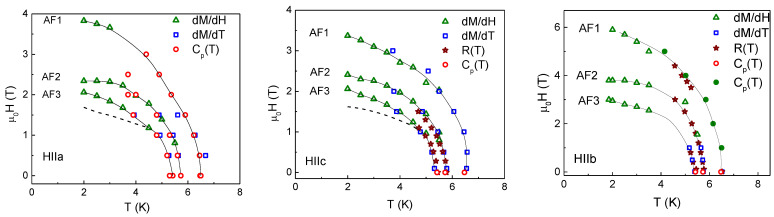
Magnetic phase diagrams of TbTe_3_ in a magnetic field oriented along *a*, *b*, and *c* axes. The dash lines mark positions of additional, sometimes smeared anomalies. Solid circles are taken from Ref. [14].

## Data Availability

Data are available on request.

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
