# Peer review of "Magnetic Phase Diagram of van der Waals Antiferromagnet TbTe3"

_materials, 2022, doi:10.3390/ma15248772_

Round 1

Reviewer 1 Report

In this work, the authors report on Transition metal-based van der Waals compounds in TbTe3 through Combining thermodynamic, specific heat Cp and magnetization M, and transport, resistance R, measurements.

The application of magnetic field shifts these boundaries to lower temperatures in parallel. In thermodynamic properties, the anomalies an TN3 and TN2 are seen as the sharp signatures of the first order transitions, which can be due to giant magnetostriction of the rare-earth compounds.

The presented data seems to be of good quality, the experiments were well done,
I would recommend it for publications with a minor revision and to pay attention to the following comments before publications.
1.             The citation of previous literature in the introduction needs to be arranged from the way the authors use their citations. 2.             The authors should carefully exam the obtained stoichiometry, which can be confirmed via x-ray fluorescence microanalysis. Is TbTe3 exactly 1-3 or something else?  
More details on the description of the atomic structure as well as on stoichiometry should be well
presented.

Author Response

Reviewer 1.

In this work, the authors report on Transition metal-based van der Waals compounds in TbTe3 through Combining thermodynamic, specific heat Cp and magnetization M, and transport, resistance R, measurements. The application of magnetic field shifts these boundaries to lower temperatures in parallel. In thermodynamic properties, the anomalies an TN3 and TN2 are seen as the sharp signatures of the first order transitions, which can be due to giant magnetostriction of the rare-earth compounds. The presented data seems to be of good quality, the experiments were well done, I would recommend it for publications with a minor revision and to pay attention to the following comments before publications.

  1. The citation of previous literature in the introduction needs to be arranged from the way the authors use their citations. 

The references in the introduction appear in correct order.

  1. The authors should carefully exam the obtained stoichiometry, which can be confirmed via x-ray fluorescence microanalysis. Is TbTe3 exactly 1-3 or something else?  More details on the description of the atomic structure as well as on stoichiometry should be well presented.

The quality of selected crystals and the spatial arrangement of crystallographic axis were controlled using X-ray diffraction techniques.

Reviewer 2 Report

This manuscript is devoted to the study and construction of magnetic phase diagram for TbTe3 van der Waals antiferromagnet. The magnetic phase transitions were studied by using the specific heat, magnetization and resistance measurements. The obtained results are very interesting and useful for researchers working in area.

Author Response

Reviewer 2.

This manuscript is devoted to the study and construction of magnetic phase diagram for TbTe3 van der Waals antiferromagnet. The magnetic phase transitions were studied by using the specific heat, magnetization and resistance measurements. The obtained results are very interesting and useful for researchers working in area.

Thanks for the positive estimation of our research.

Reviewer 3 Report

The authors report the magnetic phase diagram of TbTe3 by transport and basic characterization methods. Recently there is a growing interest in the field of ReTe3 due to the novel properties of charge density wave and magnetism. I think the authors’ work is scientifically sound and therefore suggest the publication.

Here are some comments:

1.      The writing for abstracts needs to be improved. For example, in line 17, ‘basing on’ should be ‘based on’. In line 15, transport and resistance measurements are repetitive here.

2.      The authors should provide more details on how they prepare the sample for transport measurements, like how they do the contacts.

3.      It is hard for me to identify the small jumps in Figure 4c.

4.      Does the author have R vs T data at lower temperatures, say down to 2K?

Author Response

Reviewer 3.

The authors report the magnetic phase diagram of TbTe3 by transport and basic characterization methods. Recently there is a growing interest in the field of ReTe3 due to the novel properties of charge density wave and magnetism. I think the authors’ work is scientifically sound and therefore suggest the publication.

Here are some comments:

  1. The writing for abstracts needs to be improved. For example, in line 17, ‘basing on’ should be ‘based on’. In line 15, transport and resistance measurements are repetitive here.

Corrected. It is written transport, i.e., resistance. We prefer exact definition since other transport measurements are possible: thermal conductivity, Hall effect, etc.

  1. The authors should provide more details on how they prepare the sample for transport measurements, like how they do the contacts.

From untwinned single crystals with a thickness typically 0.5–1.0 μm, we cut bridges with a width 50–100 μm oriented along c- and a-axis. Measurements of electrical resistivity have been performed with a conventional 4-probe configuration. For contacts preparation we used gold evaporation and cold soldering by In as it was done in Ref. [Solid State Communications,188 (2014) 67].

  1. It is hard for me to identify the small jumps in Figure 4c.

      Yes, these jumps are smaller at H//b axis as compared to jumps at H//a and H//c axes. Nevertheless, they can be seen in Figure 4c and in derivatives of magnetization.

  1. Does the author have R vs T data at lower temperatures, say down to 2K?

      No. The measurements were done down to 4.2 K.

Reviewer 4 Report

The authors show the data of numerous measurements for thermodynamic
characterization and transport properties on TbTe3. They argue that their
data is in accordance with three phase transitions related to magnetic long-range order and a CDW. The data appears to be original and the scientific work is solid. The presentation, however, can still be improved along the suggested lines below.

    General points:

    (i) For the reader it would be very beneficial if you included a figures showing the three orderings suggested for TbTe3. From which sources are these ordering patterns known? Please cite the relevant inelastic neutron scattering studies.

    (ii) What is the spin of Tb in this compound?
    Is it 5\hbar as the magnetic moment suggests?

    line 16: It remains unclear what is commensurate and what is commensurate, magnetization, charges? Please formulate unambiguously.

    line 17: basing -> based

    line 49: trittelurides -> tritellurides

    line 63: aimed at determination of ...

    line 102: please add a sentence and a reference on what Fisher's specific heat is and why it is interesting to compare it to Cp.

    line 113: "multiple kinks" is a bit vague. Do you see two or three very steep rises in each panel? They are not *at* the transition temperatures - are the shifts due to the finite fields?

    line 124: Why does the data suggest that the moments are ordered in the ac plane? Please explain this more carefully since it is a relevant result of yours.

    line 130: Why is the material insulating in spite of the large magnetic moment? Please add two sentences stating the present understanding of this issue.

    line 136 and 137: I got confused: What is the difference between a double layer and a bilayer? Please introduce and illustrate your terminology clearly in Fig. 1.

    line 165: Please explain: Why is giant magnetorestriction an origin of first order transitions?

    line 167: This sentence starts strange; should it read "It is not always ..."?

    All in all, I will be able to recommend publication once the above points will have been improved.

Author Response

Reviewer 4.

The authors show the data of numerous measurements for thermodynamic characterization and transport properties on TbTe3. They argue that their data is in accordance with three phase transitions related to magnetic long-range order and a CDW. The data appears to be original and the scientific work is solid. The presentation, however, can still be improved along the suggested lines below.

    General points:

    (i) For the reader it would be very beneficial if you included a figures showing the three orderings suggested for TbTe3. From which sources are these ordering patterns known? Please cite the relevant inelastic neutron scattering studies.

These three temperatures of magnetic ordering are clearly seen as peaks in temperature dependencies of specific heat and Fisher’s specific heat shown in right panel of Fig. 2. It is written in the text that “In accordance with neutron diffraction data [9] the ordering vectors of antiferromagnetic structures below TN1 (AF1) and TN2 (AF2) are equal to qm1 = (1/2,1/2,0) and qm2 = (0,0,1/2).” Below TN3, the magnetic structure is incommensurate with crystal lattice.

    (ii) What is the spin of Tb in this compound?  Is it 5\hbar as the magnetic moment suggests?

The spin moment of Tb is 3, but due to the spin-orbit coupling its full moment is 6. It is written in the text that “The value of Curie constant allows estimating the effective magnetic moment of Tb3+ ions according to the ratio 8C = µeff2. It gives the µeff = 9.92 µB which is in a good correspondence with tabular magnetic moment of Tb3+ ions µcalc = 9.7 µB.”

    line 16: It remains unclear what is commensurate and what is commensurate, magnetization, charges? Please formulate unambiguously.

In TbTe3, both magnetism and charge are incommensurate with the lattice. It is written in the text that “Slight difference in a and c lattice constants originates from the formation of charge density wave with propagation vector qc = (0, 0, 0.296) at Tc = 330 K [10].” In this reference it is stated also that in all three magnetically ordered phases, incommensurate modulations are present. The magnetic propagation vector of the (0, 0, 0.5±δ)- type observed just below TN1 turns out to be locked-in below TN3 in the simple antiferromagnetic position (0, 0, 0.5). However, the magnetic Bragg peaks observed below TN1 near the position (0, 0, 0.24) stabilize at low temperature at the incommensurate position (0, 0 + δ, 0.21). This effect is possibly due to the incommensurate lattice modulation present in TbTe3 in the CDW state.  Since we do not have our own neutron scattering data on TbTe3, we prefer to cite Ref. 10. Necessary remark is added to the text.

    line 17: basing -> based

Corrected

    line 49: trittelurides -> tritellurides

            Corrected

    line 63: aimed at determination of ...

            Corrected

    line 102: please add a sentence and a reference on what Fisher's specific heat is and why it is interesting to compare it to Cp.

Added to the text: This function is closely similar to the magnetic specific heat in the region of transition [13]. These three anomalies are seen also in Cp vs. T curve which mainly consists of the contribution from the magnetic subsystem in the region of transition sharpest of them is detected at TN2.

    line 113: "multiple kinks" is a bit vague. Do you see two or three very steep rises in each panel? They are not *at* the transition temperatures - are the shifts due to the finite fields?

We detect two sharp peaks at TN1 and TN2 and a shoulder at TN3 in all temperature dependencies seen more clearly in Fisher’s specific heat in the right panel of Fig. 2. These anomalies are suppressed by external magnetic field.

    line 124: Why does the data suggest that the moments are ordered in the ac plane? Please explain this more carefully since it is a relevant result of yours.

We added to the text: The magnetization curves measured for H applied along the a and c axes are close to zero in weak magnetic fields and evidence a spin – flip transition at µ0H = 2.33 T. While M(H) curve measured along the b axis grows almost linear in external magnetic field and evidences a sequence of small jumps at critical fields.

line 130: Why is the material insulating in spite of the large magnetic moment? Please add two sentences stating the present understanding of this issue.

This is misunderstanding. This material is a metal.

    line 136 and 137: I got confused: What is the difference between a double layer and a bilayer? Please introduce and illustrate your terminology clearly in Fig. 1.

Terbium bilayer is substituted by Tb – Te bilayer. The description of the structure is corrected in the caption to Fig. 1.

    line 165: Please explain: Why is giant magnetostriction an origin of first order transitions?

Giant magnetostriction of the rare-earth compounds results in sizable changes in crystal lattice parameters which transform purely magnetic transitions of second order into magnetostructural phase transitions of first order. It is accompanied by a change in the shape of specific heat anomaly of lambda-type to symmetric sharp peak. Necessary explanations are added to the text.

    line 167: This sentence starts strange; should it read "It is not always ..."?    

            Yes, corrected.

    All in all, I will be able to recommend publication once the above points will have been improved.

Reviewer 5 Report

The present manuscript reports on the magnetic, resistivity and heat capacity measurements in TbTe3 single crystals. The main result consists in establishing the magnetic H-T phase diagram of the antiferromagnetic-like transitions taking place below 10 K along the three crystallographic axes.

In general, data collection and data analysis are sound. The heat capacity measurements are most convincing in supporting the magnetic phase diagram.

The text is more a technical report than a scientific article as it presents the data without attempting at broadening the discussion beyond what is presently observed in TbTe3. Yet, I recommend considering this manuscript for publication in Materials due to the systematic character of this work.

I however recommend the authors to pay attention to the following two points (minor corrections not calling for further review).

- The heat capacity measurements are not described in the method section.

- Invoking first order transition in the conclusion at l165 is not supported in the main text. The sharpness is not a sufficient criterion to distinguish first from second order transitions. More data such as a determination of the thermal hysteresis or a higher temperature resolution during Cp measurements (lambda-like vs symmetrical peak distinction) should be presented for a more accurate determination of the Ehrenfest order of the transitions. Alternatively, L165 and 166 could be deleted since they are not essential to the manuscript.

Author Response

Reviewer 5.

The present manuscript reports on the magnetic, resistivity and heat capacity measurements in TbTe3 single crystals. The main result consists in establishing the magnetic H-T phase diagram of the antiferromagnetic-like transitions taking place below 10 K along the three crystallographic axes.

In general, data collection and data analysis are sound. The heat capacity measurements are most convincing in supporting the magnetic phase diagram.

The text is more a technical report than a scientific article as it presents the data without attempting at broadening the discussion beyond what is presently observed in TbTe3. Yet, I recommend considering this manuscript for publication in Materials due to the systematic character of this work.

I however recommend the authors to pay attention to the following two points (minor corrections not calling for further review).

- The heat capacity measurements are not described in the method section.

The details of measurements of specific heat are presented in Experimental section, line 78.

- Invoking first order transition in the conclusion at l165 is not supported in the main text. The sharpness is not a sufficient criterion to distinguish first from second order transitions. More data such as a determination of the thermal hysteresis or a higher temperature resolution during Cp measurements (lambda-like vs symmetrical peak distinction) should be presented for a more accurate determination of the Ehrenfest order of the transitions. Alternatively, L165 and 166 could be deleted since they are not essential to the manuscript.

Giant magnetostriction of the rare-earth compounds transforms λ-type anomaly inherent for a second order phase transition of purely magnetic origin to symmetric sharp peak at magnetostructural phase transition. This sentence is added to the text.